# Contextualizing the Role of Osteopontin in the Inflammatory Responses of Alzheimer’s Disease

**DOI:** 10.3390/biomedicines11123232

**Published:** 2023-12-06

**Authors:** Roshni C. Lalwani, Claude-Henry Volmar, Claes Wahlestedt, Keith A. Webster, Lina A. Shehadeh

**Affiliations:** 1Interdisciplinary Stem Cell Institute, Leonard M. Miller School of Medicine, University of Miami, Miami, FL 33136, USA; rcl88@med.miami.edu; 2Department of Psychiatry, Leonard M. Miller School of Medicine, University of Miami, Miami, FL 33136, USA; cvolmar@med.miami.edu (C.-H.V.); cwahlestedt@med.miami.edu (C.W.); 3Center for Therapeutic Innovation, Leonard M. Miller School of Medicine, University of Miami, Miami, FL 33136, USA; 4Integene International Holdings, LLC, Miami, FL 33137, USA; kwebster@med.miami.edu; 5Department of Ophthalmology, Baylor College of Medicine, Houston, TX 77030, USA; 6Everglades BioPharma, Houston, TX 77098, USA; 7Department of Medicine, Leonard M. Miller School of Medicine, University of Miami, Miami, FL 33136, USA

**Keywords:** Alzheimer’s disease, osteopontin, Spp1, microglia

## Abstract

Alzheimer’s disease (AD) is characterized by progressive accumulations of extracellular amyloid-beta (Aβ) aggregates from soluble oligomers to insoluble plaques and hyperphosphorylated intraneuronal tau, also from soluble oligomers to insoluble neurofibrillary tangles (NFTs). Tau and Aβ complexes spread from the entorhinal cortex of the brain to interconnected regions, where they bind pattern recognition receptors on microglia and astroglia to trigger inflammation and neurotoxicity that ultimately lead to neurodegeneration and clinical AD. Systemic inflammation is initiated by Aβ’s egress into the circulation, which may be secondary to microglial activation and can confer both destructive and reparative actions. Microglial activation pathways and downstream drivers of Aβ/NFT neurotoxicity, including inflammatory regulators, are primary targets for AD therapy. Osteopontin (OPN), an inflammatory cytokine and biomarker of AD, is implicated in Aβ clearance and toxicity, microglial activation, and inflammation, and is considered to be a potential therapeutic target. Here, using the most relevant works from the literature, we review and contextualize the evidence for a central role of OPN and associated inflammation in AD.

## 1. Introduction

Alzheimer’s disease (AD), the fifth leading global cause of death, is a neurodegenerative condition that, along with subcortical vascular cognitive impairment (SVCI), is responsible for 75% of all dementia cases [1,2,3]. Current estimates place the global population of dementia at 57.4 million, and this is predicted to grow to 152.8 million by the year 2050 in parallel with aging populations, becoming a correspondingly greater public health concern [4,5]. There is no cure for AD, and its progressive course, along with the limited efficacy of current disease-modifying treatments, severely impacts the quality of life of those who are afflicted, with potentially devastating familial, social, and economic consequences. The clinical progression of AD is marked by episodic memory loss and impaired learning, followed by heterogenous changes in executive and visuospatial function [3]. The 2020 Lancet Commission on dementia prevention, intervention, and care identified a set of modifiable risk factors for late onset AD that include, but are not necessarily limited to, hypertension, smoking, obesity, physical inactivity, diabetes, alcohol, traumatic brain injury, and air pollution, and suggest that interventions that target such factors could prevent up to 40% of dementia prevalence [6,7,8]. Whereas age is the main risk factor for AD, more than 50 genetic risk loci are known, which include genes that regulate lipid homeostasis and inflammation, [9,10,11,12] with the strongest association being that with the ε4 allele of *ApoE* (*ApoE4*) [13,14,15]. In keeping with the genetic risk factors that predispose to AD, inflammation and oxidative stress feature prominently in such modifiable lifestyle factors.

AD remains an urgent unmet clinical need, with multiple ongoing clinical trials and an increasingly diverse range of potential therapeutic targets in the pipeline. One of these is osteopontin (OPN), an inflammatory cytokine and diagnostic marker of AD that is implicated in the regulation of activated microglia associated with AD. Here, we review current knowledge and recent advances in the involvement of OPN with established neuropathological pathways of AD. The literature supports central albeit complex roles of OPN in regulating autophagic and inflammatory responses to amyloid-beta (Aβ) and tau accumulations in the brain that coincide with AD progression and represent possible new therapeutic targets. However, such therapeutic application is tempered by complex interactions of OPN with both pro- and anti-inflammatory microglial subtypes as well as other cell types, and the possible switching of OPN actions between neurodegeneration and neuroprotection in a disease-stage-dependent manner. These possibilities are discussed, and our overall assessment supports a cautious approach to the clinical translation of anti-OPN strategies.

## 2. Aβ/NFT Pathway

Despite the high prevalence of AD that has driven intense clinical and translational research efforts over the past 30 years, the complex pathophysiology remains poorly understood. The amyloid-beta (Aβ) and tau pathways proposed in the late 1980s and early 1990s have become hallmarks of AD [16]. The Aβ component involves the deposition of extracellular senile plaques of Aβ, products of cleavage of a larger amyloid precursor protein (APP) by β- and γ-secretases [16,17,18] (see Figure 1), that is naturally produced in the brain by neurons, vascular and blood cells, and astrocytes. APP is involved in multiple neuronal functions including neurite outgrowth and axonal guidance, the regulation of synaptic function and plasticity, early nervous system development, and neuroprotection [19]. Aβ aggregates progress from soluble oligomers and protofibrils to insoluble fibrils and plaques, all of which may be neurotoxic [20]. The second component involves accumulations of abnormally phosphorylated tau, a protein codified by the alternative splicing of the microtubule-associate protein tau (*MAPT*) gene that is enriched in axons of mature neurons, where it stabilizes the microtubules required for neuronal transport and structure. Hyperphosphorylated tau with a reduced affinity for tubulin becomes detached from microtubules and aggregates into paired helical filaments that spread and form cytoplasmic neurofibrillary tangles (NFTs) within neurons [20,21]. Neuropathological evidence indicates that increased levels of intracellular Aβ, parenchymal Aβ plaques, and NFTs are first seen in neurons of the entorhinal cortex (EC) [22,23,24]. Indeed, earlier work indicated that EC LII-neurons may be the first cortical neurons to degenerate in the course of AD [25,26]. Using molecular PET scanning in longitudinal clinical studies of subjects with varying Aβ burden and AD severity, Sanchez et al. described tau deposition in the medial temporal lobe of subjects many years prior to Aβ [27]. Tauopathy was initially detected in clinically normal people near the rhinal sulcus, and spread in association with Aβ to the neocortex of the temporal lobe and then to extratemporal regions. Pathogenically activated Aβ proteins and NFTs interact with neurons to synergistically disrupt synapses and neuronal networks [18,20,28,29]. Neuropathological, genetic, and in vivo biomarker-based evidence from animal models and humans indicate a latency period wherein accumulations of Aβ precede the spreading of plaques and NFTs, neuronal loss, and clinical manifestations of AD by 20–30 years [30,31,32,33,34]. However, increased deposits of pre-fibrillar tau and soluble Aβ show a strong correlation with cognitive decline, suggesting that substantial toxic assemblies of Aβ-peptides and tau exist outside of visible deposits [35,36]. During the latent period, physiologic responses to soluble and insoluble Aβ aggregates and NFTs within AD brains include the activation of lysosomal/endolysosomal systems and increased numbers of autophagosomes that may help to maintain homeostasis, mitigate neurodegeneration, and precede the complex cellular innate immune responses that characterize neurodegeneration and clinical decline [29,37,38].

The determination of the causes of Aβ and NFT accumulation, spreading, activation, and the propagation of neurotoxicity are fundamental to understanding the molecular etiologies of both early and late-onset AD and to identify new therapeutic targets. Tge accumulation of Aβ involves both the overproduction and blocked clearance of pathological APP fragments with the generation of more toxic isoforms such as Aβ42, a major component of both Aβ aggregates and NFTs that is prone to misfolding and aggregation. β- and γ-secretases cut APP into Aβ fragments of different lengths, mainly of 40 and 42 amino acids [39]. High ratios of Aβ42 versus Aβ40 are associated with increased tau phosphorylation and exacerbated dissociation from microfilaments [40]. Toxic soluble Aβ oligomers (AβOs) form before Aβ plaques and may be responsible for the early cognitive decline of AD patients [41]. Accumulations of Aβ and hyperphosphorylated tau synergistically trigger prion-like spreading, wherein abnormally folded proteins migrate to adjacent neurons and form seeds that convert normal proteins into pathological forms [20,42,43]. The combined effects of accumulation and dispersal drive the spread of incipient Aβ extracellular plaque and tau/NFT aggregates to interconnected areas throughout the brain over time [16,20,33,44,45,46,47]. Secondary to the accumulation and spread of Aβ/NFT aggregates, much evidence supports roles for lipid dysregulation and inflammation as driving the pathological phenotype and progression to clinical AD [48]. Contributions of ApoE4 to AD progression are well established, and recent work indicates important roles for the low-density lipoprotein receptor (LDLR) and LDLR-related protein 1 (LRP1) in regulating tau uptake and spread, establishing important roles for lipid regulators in AD progression and possible therapeutic targets [49,50,51]. By polarizing microglia towards an M2 phenotype with anti-inflammatory and phagocytic properties, OPN has also been linked with the clearance of Aβ plaques as well as neuroprotection by delivering pro-survival, anti-apoptotic signals [52,53,54].

## 3. Etiology: Molecular Genetic and Epigenetic

Genetic evidence strongly supports the hypothesis that Aβ/tau initiates the early-onset form of AD. Large-scale genetic analyses of AD pedigrees identify highly penetrant mutations in APP, and genes encoding the subunits of the γ-secretase APP cleavage enzymes presenilin 1 and 2, as causal for dominantly inherited early-onset AD [55,56,57,58]. Mutations in the microtubule-associate protein tau (*MAPT*) gene that confer the abnormal production and aggregation of tau leading to NFT are also dominantly inherited and cause early-onset frontotemporal dementia [58]. The results are consistent with Aβ/NFT as diagnostic for AD. Whereas no causal genetic mutations are known for late-onset AD, that does not follow Mendelian inheritance, significant heritability is apparent, with evidence that the risk of AD may be as much as 80% dependent on heritable factors [9,10,11,59]. The expression of the ε4 allele of the *ApoE* gene (*ApoE4*) confers the highest risk for late-onset AD [10]. However, large-scale genome-wide association studies identified multiple additional critical genetic risk factors associated with more than 50 susceptibility gene loci including *CLU*, *PICALM*, *CR1*, *BIN1*, *MS4A*, *CD2AP*, *CD33*, *EPHA1*, and *ABCA7* [60,61,62,63]. Many of these genes are linked directly to Aβ and tau homeostasis and include genes involved in inflammation and immune response pathways, cellular trafficking, lipid metabolism, and ubiquitination pathways [59]. Such studies revealed strong associations of variants in genes for immune receptors, *TREM2* (triggering receptor expressed on myeloid cells 2) [64], and *CD33* [65,66] with late AD. The results further corroborate the links between activated Aβ/tau, inflammation, and AD progression. By blocking the clearance of Aβ, vascular dysfunction is also implicated in AD progression and may be the mechanism whereby cerebral small vessel disease (CSVD) exacerbates AD [67,68,69,70]. Through its differential regulation of integrin-mediated signaling and neuroinflammation, OPN has long been known to have central roles in inflammatory CNS diseases including MS and AD, driving both neurodegenerative and protective pathways (reviewed in [54]).

## 4. ApoE and LDLR

ApoE, the primary lipid and cholesterol transporters in the central nervous system, facilitate the transport of lipids by binding to LDLR and LRP1 [71]. ApoE is abundantly expressed in astrocytes, microglia, vascular mural cells, and choroid plexus cells, where it differentially modulates neuronal intracellular signaling pathways, including synaptic homeostasis, glucose metabolism, and cerebrovascular function [72]. Of the three ApoE isoforms expressed in the human brain, the presence of ApoE4 correlates with rapid AD onset and neurodegenerative decline and poses the highest genetic risk for AD, whereas ApoE2 is protective [13,14,15]. Amino acid substitutions on the N- and C-terminus of the proteins that, respectively, contain LDLR and lipid binding sites of ApoE proteins are responsible for the differences [73]. A large body of evidence indicates that ApoE4 binds more strongly to Aβ, promoting the aggregation, stabilization, and deposition of fibrillar plaques and reducing clearance, whereas ApoE2 binds weakly, reduces Aβ aggregates and NFTs, and enhances Aβ clearance by microglia [74,75]. Compelling evidence from patients and mouse models also indicates that ApoE4 drives tau accumulation and redistribution to neuronal cell bodies in a manner that promotes pathological microglial activation and neuroinflammation. Whereas these effects appear to be mediated by the direct interaction of ApoE with tau independently of Aβ, the precise mechanisms are unclear [48,76,77,78]. The results are consistent with roles for ApoE subtypes in the binding and/or activation of microglia during AD progression.

Only non-lipidated ApoE binds Aβ plaque. Reducing the total ApoE levels in the brain or increasing its lipidation state markedly reduced the Aβ plaque burden and associated inflammatory phenotypes in mouse AD models [75,79,80,81,82]. Such restricted control of the Aβ/NFT burden to non-lipidated ApoE is also consistent with the established roles of the LDLR and LRP1 in regulating ApoE and Aβ plaque. For example, the overexpression of LDLR in APP/PS1 transgenic mice (ADtg) that reproduce AD features robustly and age-dependently markedly reduced ApoE, suppressed microglial activation, and ameliorated neurodegeneration [51]. Similar regulation was conferred by the expression of the ATP-binding cassette transporter A1 (ABCA1) gene, the principal vehicle for lipid transfer to ApoE and another genetic AD marker. Therapeutic benefit can be achieved in mouse ADtg models by reducing ApoE with antibodies, antisense oligonucleotides or ApoE siRNAs, or by engineering the overexpression of LDLR and/or LRP1, and can represent a promising clinical strategy [49,51,82,83,84,85].

## 5. Inflammation

Neuronal microgliosis (migration of local macrophages, astrocytes, and microglia) constitutes the principal inflammatory response to Aβ plaque deposition and correlates closely with the onset of brain cell damage and cognitive decline [86,87]. However, mouse models and patient studies also implicate significant peripheral immune cell infiltration of the brain parenchyma during different stages of AD [88]. Systemic inflammation driven by Aβ egress into the circulation via two-way transport systems such as the LDLR associated factors that circumvent the blood–brain barrier (BBB) [89,90] may contribute to AD progression by interfering with the clearance of Aβ by microglia, triggering tau hyperphosphorylation and the spread of NFT, as well as by promoting BBB breakdown [29,91,92,93,94]. Systemic myeloid cells and inflammatory cytokines can access the brain via humoral and vascular signaling pathways and confer both destructive and reparative actions [94,95,96]. However, during active AD inflammation, Aß plaques and regions of the brain with high tau accumulation are frequently found surrounded by CD11b+, Iba1+, and CD45+ immune cells that originate mostly if not entirely from microglia [51,97,98,99]. The primary function of such microglial migration is to repair and clear Aß plaque through phagocytosis and autophagy, but in the context of late-stage symptomatic AD, effective clearance fails. The reasons for failure are not understood but presumably involve overwhelmed clearance systems, due in part to the relatively late appearance of phagocytic/anti-inflammatory microglia (see below), inhibitory cytokines, and adaptive responses to the long latent period of Aβ/NFT evolution, and the continued presence of toxic AβOs [41,99].

Activated microglia can be neurotrophic or neurotoxic depending on the activating stimuli, immune milieu, and surrounding environment [48,100,101]. M1 or inflammatory microglia are usually activated via Toll-like receptors and γ-interferon signaling, producing pro-inflammatory cytokines such as interleukins (IL-1β) and tumor necrosis factor alpha (TNF-a) and chemokines, and expressing NADPH oxidase and matrix metalloproteinases [96,98]. Anti-inflammatory M2 microglia are neuroprotective, secrete growth factors, and promote the release of anti-inflammatory cytokines (reviewed in [3,102]). OPN interacts with both M1- and M2-activated microglia, showing synergistic detrimental and protective functions, respectively [52,54]. Studies in human patient and ADtg mice identified a protective disease-associated type of microglia (DAM) with an increased expression of lipid metabolism and phagocytic-related genes (Trem2, Tyrobp, Cst7, CD9) and anti-inflammation properties [103]. DAMs were reported to be associated with phagocytic Aβ clearance particles and generated at advanced stages of AD progression, such that they may determine the time of onset and the rate of disease progression. Other studies identified microglial TAM receptor tyrosine kinases Axl and Mertk, as necessary for microglia to bind and phagocytose Aβ plaque [104], but TAM-deficient ADtg mice developed lower Aβ high-density plaque loads compared with WT ADtg mice with normal microglial TAM. This unexpected and confusing result suggested to the authors that TAM-driven microglial phagocytosis of Aβ does not inhibit but rather promotes dense-core plaque development via pathways that may involve the lysosomal and microglial re-deposition of high-density Aβ material that was previously phagocytosed by exocytosis and/or microglial death pathways [105]. Yet, other studies identified an APOE-TREM2 pathway that mediated a switch of microglia from a homeostatic to a neurodegenerative phenotype that was initiated after the phagocytosis of apoptotic neurons [106]. Consistent with these works, molecular and transcriptome studies reveal the presence of multiple human and mouse microglial subsets involved in homeostasis, proliferation, interferon response, and antigen presentation that include subsets expressing enriched neurodegenerative disease genes related to AD [107].

Microglia are increasingly understood to play flexible and sometimes opposing roles in AD pathogenesis by eliminating toxic Aβ aggregates, enhancing neuronal plasticity and conferring anti-inflammatory signals, or by producing proinflammatory cytokines, ROS, and neurotoxicity [108]. Microglia express cell-surface receptors that differentially activate innate immune responses and mediate protective and destructive responses to Aβ aggregates. For example, different studies implicate CD36, CD33, and Toll-like Receptor 4 (TLR-4) in mediating inflammatory cytokine release in response to damage-associated molecular patterns (DAMPs). CD36, a recognized marker of AD, binds cholesterol, modulates innate immunity, and protects against inflammation and oxidative stress [109,110,111]. CD33 regulates innate immunity by binding sialic acids of glycoproteins and glycolipids and immunoreceptor tyrosine-based inhibition motifs, thereby restricting immune responses [110]. TLR-4 regulates both innate and adaptive immune responses through signaling cascades that lead to the upregulation of cytokines, chemokines, growth factors, and other inflammatory mediators. TLR4 binds Aβ aggregates and activates pro-inflammatory microglia with increased phagocytosis and cytokine production [108].

Investigations of the roles of microglia and inflammatory mediators on the deposition and clearance of Aβ plaques are mixed and complex, with evidence for the amelioration and exacerbation of aggregates. For example, the gene-therapy-mediated suppression of brain anti-inflammatory interleukin 10 (IL10) in ADtg mice was shown to increase Aβ42 aggregates and the plaque burden [112], whereas the KO of IL10 in the same ADtg model had the opposite effect, reducing Aβ aggregates [113]. In one study, a depletion of microglia from ADtg mice did not affect Aβ plaque deposition or clearance [114]. The results indicate the overriding effect of disease stage and physiologic/pathological context; subsets of microglia with different receptors and inflammation mediation pathways are activated at different stages of the cellular response to AD and can confer benefit or injury in stage-dependent and quantitative manners. Molecular genetic analyses of age-related gene expression in AD brains indicate that aging predisposes the brain to inflammatory processes consistent with an age-dependent selection of microglial subsets [115]. Taken together, the results suggest that AD pathology is defined at least in part by age-dependent changes in the activation of myeloid cells and microglia and the context-dependent balance of pro versus anti-inflammation phenotypes of infiltrating myeloid and resident microglia [102].

## 6. Osteopontin

OPN is a negatively charged acidic phosphoprotein that circulates as a pro-inflammatory cytokine [116]. Variously referred to as early T-cell activation protein (ETA-1), bone sialoprotein (BSP-1), and secreted phosphoprotein (SPP-1), OPN is multifunctional, with different functional domains exposed through thrombin or metalloprotease cleavage [3,54] (see Figure 2). Thrombin cleavage generates an N-terminal OPN fragment that binds integrins and mediates cell adhesion and migration, bone marrow immune responses, and inflammation [3]. OPN is found in the bone matrix, kidney, placenta, and blood vessels and contributes to multiple physiologic and pathologic processes, including bone mineralization, oxidative stress, remyelination, inflammation, immunity, and wound healing [116,117]. OPN, secreted by T cells and tissue-resident macrophages, has been shown to regulate macrophage functions, including migration, reparative and degenerative phagocytosis, and chemotaxis in different peripheral tissues [118,119,120]. In the brain, OPN expression is precisely regulated in a spatiotemporal and cell-type specific manner, depending on context, age, and brain region, where it regulates responses of immune cells to injury [95,121]. OPN levels may be elevated in the plasma, urine, cerebrospinal fluid (CSF), and brain of subjects with neurodegenerative disease, including AD [3,95,122,123,124,125], multiple sclerosis (MS) [116,122,126], Parkinson’s disease [116,122], amyotrophic lateral sclerosis [3,127], HIV-associated neurocognitive disorder [4,128], and mild cognitive impairment (MCI) [129]. Increased OPN after neuronal damage coincides with glial cell recruitment and the influx of inflammatory cytokines [130]. In a manner that parallels the context-dependent, opposing actions of microglial subtypes, different studies report that OPN can promote inflammatory damage or neuroprotection and repair [54].

**Figure 2 biomedicines-11-03232-f002:**
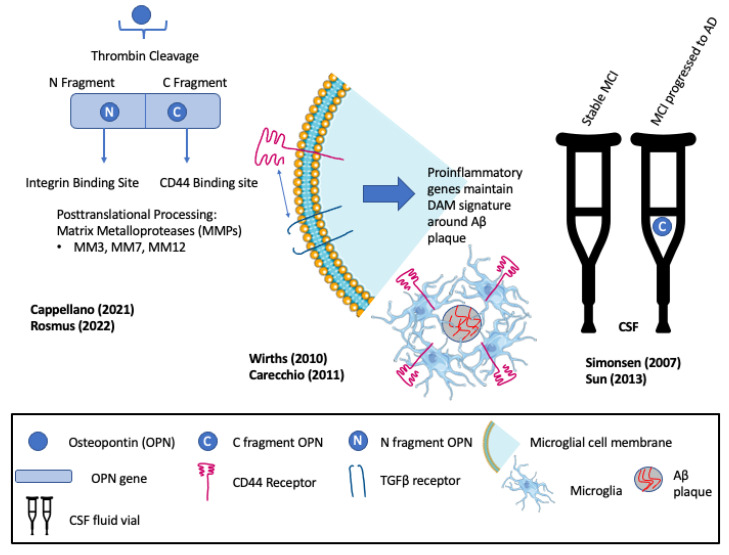
Multifunctionality of osteopontin (OPN) and its roles in recruiting and maintaining disease-associated microglia (DAM). Elevated levels of the C-fragment of OPN in cerebrospinal fluid (CSF) mark the progression of patients with mild cognitive impairment (MCI) to clinical Alzheimer’s disease (AD). Rosmus et al., Wirths et al., Cappellano et al., Sun et al., Simonsen et al., Carecchio et al. [3,21,54,116,126,130].

Early evidence for roles of OPN in neurodegenerative disease reported an association with severe intracranial calcification, wherein calcification within the basal ganglia and cerebellum, including type 3 capillary calcospherites, express elevated OPN [117]. Similarly, OPN is found to be associated with the TGFβ signaling factor phosphor-SMAD2/3 and collagen-1 in calcified vessel walls of the cerebral cortex of patients with cerebral amyloid angiopathy [131]. Subsequently, increased OPN expression was described in the pyramidal neurons of the CA1 region of the hippocampus associated with Aβ plaque of symptomatic AD patients [44]. Proteomic analyses of CSF from patients with stable MCI versus MCI with AD identified the phosphorylated C-terminal thrombin cleavage fragment of OPN as a biomarker of MCI that progressed to AD [126]. Positive correlations between the levels of the OPN C-terminal fragment in the CSF and elevated inflammatory markers suggested a mechanistic link between OPN and increased inflammation and gliosis as MCI progressed to AD [126] (see Figure 2). Consistent with this, increased levels of OPN in the CSF were found to distinguish AD patients from those with frontotemporal dementia (FTD), and correlated with the severity of cognitive impairment, consistent with OPN as a prognostic indicator of AD progression [44]. Parallel conclusions were derived from MS patients where elevations of OPN in the brain, CSF, and serum coincided with MS stage [44]. Early immunocytochemical studies that located OPN exclusively in the cytoplasm of pyramidal neurons of AD patients [44] appear to be at variance with multiple subsequent studies that locate OPN with multiple neuronal cell types, including hippocampal perivascular macrophages, astrocytes, and microglia (see below).

Elevated OPN expression in the astrocytes and microglia of ADtg mice was shown to correlate with similarly elevated levels of inflammation and oxidative stress markers [21]. OPN, GFAP, Cathepsin D, Toll-like receptors (TLRs), and TGFβ-1 correlated with the progressive age-dependent loss of CA1 pyramidal neurons, axons, and quantitative deficits in cognitive and motor performance. The results identified OPN as a marker of astrocytes and microglial activation and inflammation [21]. Consistent with this, and in agreement with the proteomic analyses described above [126], elevated OPN levels in CSF are also increased in patients with mild cognitive impairment (MCI) that progress to AD, but not in patients with stable MCI [116] (see Figure 2). Plasma and CSF OPN are currently recognized biomarkers of both AD and vascular cognitive impairment [4]. Taken together with studies that localize OPN to areas of high Aβ plaque and activated microglia, the results are consistent with diagnostic, prognostic, and functional roles for OPN in AD [116].

In support of pro-inflammatory, neurodegenerative roles of OPN, represented in Table 1 and Figure 3 and Figure 4, one study highlighted that the antibody-mediated activation of the TREM2 receptor in ADtg mice displayed enhanced survival and proliferation of microglia and ameliorated Aβ-induced pathology and the downregulation of OPN [132,133] (See Figure 3). TREM2 is a receptor for microglial lipids and a genetic risk factor for AD. The results identified OPN as a marker of inflammation that correlates positively with neurodegeneration and negatively with protective microglia migrations to sites of Aβ plaque. Consistent with this, De Schepper et al. report that OPN secreted by hippocampal perivascular macrophages (PVM), the primary responders to toxic agents and pathogens that cross the BBB, transformed microglia into phagocytic cells that engulf synapses in ADtg mice and possibly AD patient tissues [121]. OPN was found to upregulate multiple phagocytic markers including C1qa, Grn, and Ctsb in microglia associated with Aβ oligomers (see Figure 4). This study also showed that the genetic ablation of OPN results in less synaptic loss but a resilient high Aβ burden. The mechanism of increased OPN production was unclear, but may involve signals from Aβ deposited along the vasculature associated with elevated TGF-β. Because there is no obvious physiological benefit to destroying synapses, the pathway defines OPN-mediated phagocytic and endolysosomal pathways designed to eliminate Aβ plaque that become self-destructive and drivers of AD progression [121,134]. Elevated OPN is associated with other neurodegenerative disease models including ALS and may be part of a conserved molecular response to perturbed perivascular homeostasis beyond Aβ pathology [121].

**Figure 3 biomedicines-11-03232-f003:**
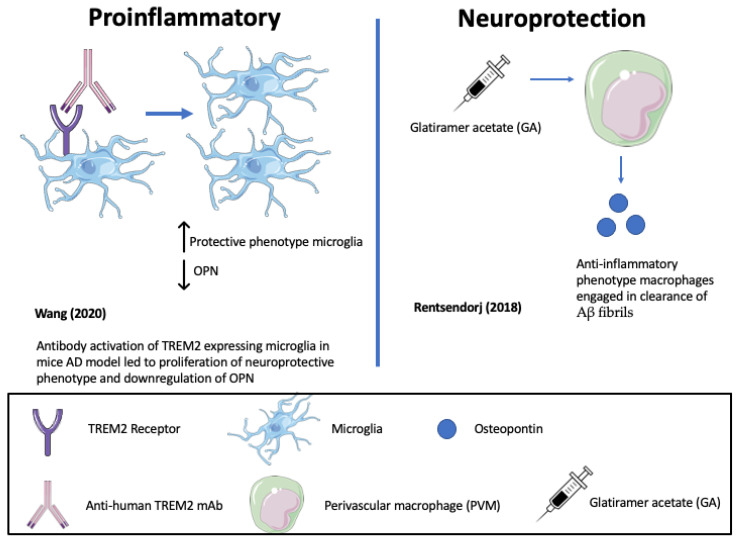
Opposing roles of osteopontin (OPN) in driving Alzheimer’s disease (AD) progression (Wang (2020) [133]; Rentsendorj (2018) [95]).

**Figure 4 biomedicines-11-03232-f004:**
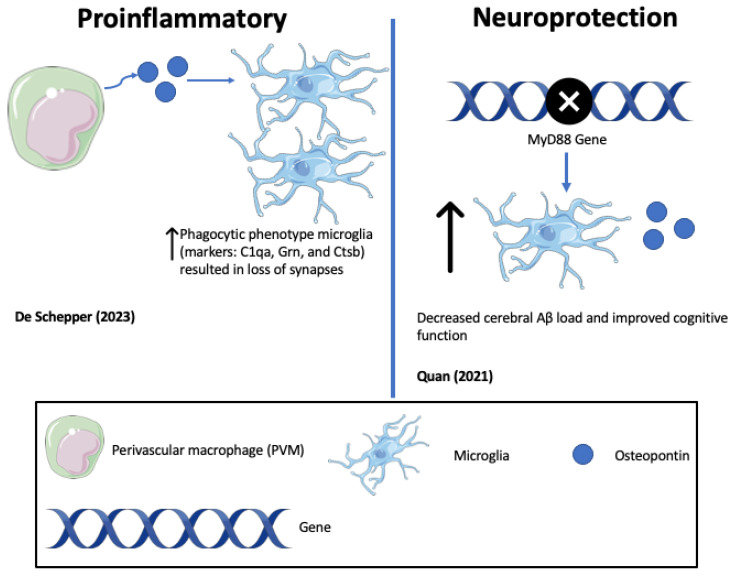
Opposing roles of osteopontin (OPN) in driving Alzheimer’s disease (AD) pathology (De Schepper (2023) [121]; Quan (2021) [135]).

Conversely, as highlighted in Table 1 and Figure 3 and Figure 4, in other mouse ADtg models and in human AD brains, elevated OPN expression correlated with protective, anti-inflammatory phenotypes and accelerated Aβ clearance [95]. When ADtg mice were injected with glatiramer acetate (GA), an immune-modulating medication used to treat MS patients, the OPN expression associated with monocytes and macrophages of systemic origin engaged in Aβ clearance and tissue repair and was found in areas of high Aβ plaque. Infiltrating OPN-expressing cells included subpopulations of CD115+CD11b+Ly6Chigh monocytes and CD11b+Ly6C+CD45high monocyte/macrophages. Similar effects were achieved by delivering peripheral blood enriched with bone marrow (BM)-derived CD115+ monocytes. Correlation matrix analyses indicated a strong linear correlation between cerebral OPN levels and macrophage infiltration, and an inverse relation between OPN and Aβ plaque burden. In vitro studies corroborated the findings, showing that GA directly upregulates OPN expression in BM-derived macrophages and promotes a phenotypic shift to phagocytic, anti-inflammatory effect with an increased uptake of Aβ fibrils [95] (see Figure 3).

Consistent with neuroprotection by elevated OPN, Quan et al. [135] reported that the heterozygous deletion of the myeloid differentiation factor 88 (MyD88) gene, an essential mediator of innate immunity, selectively in microglia of ADtg mice at a late stage of disease, decreased cerebral Aβ load and improved cognitive function, coincident with reduced microglia and inflammatory gene expression. The effects were paralleled by the up-regulation of microglial OPN and elevated LRP1 in cerebral capillaries (see Figure 4). The authors suggest that *MyD88* deficiency inhibits the activation of pro-inflammatory microglia but enhances microglial response to Aβ. Other laboratories have reported that MyD88 confers Aβ clearance and neuroprotection [136], while the deletion or disruption of *MyD88* signaling attenuates Aβ pathology [137], or is without effect [138]. The authors speculate that the responses are disease-stage-specific.

It is noteworthy that OPN C- and N-terminal fragments were previously shown to predict the presence of AD conversion in MCI patients [116,126]. Taken together, these studies suggest that the microglial expression of OPN is associated with pro-inflammatory, neurodegenerative pathways during AD, whereas OPN expressing systemic monocytes confers protective/reparative functions. Further, extracellular OPN can transform subpopulations of microglia to a phagocytic pro-inflammatory subset that engulfs and destroys synapses while BM-derived monocytes are converted to phagocytic anti-inflammatory states that engulf and remove Aβ. Therefore, the actions of OPN depend on the context, cell types, stage of disease, and prevailing biochemical interactions. Protection from AD progression in OPN-ablated ADtg mice suggests a net pathological/neurodegenerative effect. These complex properties with apparently opposing functions in different cells at disease stages make OPN a challenging target for AD therapy that will require physical and temporally restrictive exposure to modulating drugs, and/or interventions that target downstream signaling molecules.

In another approach to unravel the role of OPN in microglial activation, Qiu et al. identified a subset of CD11+ microglia that are the sole source of OPN in ADtg mice [139] (see Figure 5). The genetic deletion of OPN in ADtg mice markedly reduced proinflammatory microglia, Aβ plaques, and dystrophic neurites, while conserving cognitive function. OPN production in the periplaque areas was found to differentiate DAM into two distinct subpopulations, including a protective CD11c+OPN− subset with an increased expression of activated lysosomal markers that robustly takes up Aβ in a noninflammatory manner, and a pathogenic CD11c+OPN+ subset that produces proinflammatory cytokines including TNF-α, expresses reduced periplaque levels of TREM2, and does not ingest Aβ fibrils. Both subsets of microglia localized around plaques. The authors went on to demonstrate a threefold increase in CD11c+ OPN+ microglia in brain sections from AD patients compared with normal controls or MCI patients. In human studies, they report strong correlations between CD11c+ OPN+ microglia and CDR score, and a density of neuritic plaques and neurofibrillary tangles, highlighting the strong correlation of CD11c+ OPN+ microglia with AD severity in patients. Finally, the authors show that weekly intravenous injections of anti-OPN mAb into 6-month-old ADtg mice significantly reduced TNF-α+ CD11c+ microglia and total plaque burden after 2 months of treatment, despite having to traverse the BBB. The work may resolve some of the controversy over OPN’s contribution to microglial actions and neurodegeneration. Indeed, the results may help to resolve another quandary involving microglia that confer alternate beneficial and injurious roles in MS and wherein osteopontin has also been identified as a potential candidate for regulating the switch [140].

**Table 1 biomedicines-11-03232-t001:** Chronologic list of publications on osteopontin in Alzheimer’s disease studies.

Author	Title	Major Findings
Fujita (2003) [117]	Immunohistochemical examination on intracranial calcification in neurodegenerative diseases	Neurodegenerative diseases have increased intracranial calcification (within basal ganglia and cerebellum) wherein the type 3 capillary calcospherites express increased OPN.
Wung (2007) [44]	Increased expression of the remodeling- and tumorigenic-associated factor osteopontin in pyramidal neurons of the Alzheimer’s disease brain	Increased OPN expression in pyramidal neurons of the Ca1 region of the hippocampus are associated with Aβ plaque of symptomatic AD patients.
Simonsen (2007) [126]	Novel Panel of Cerebrospinal Fluid Biomarkers for the Prediction of Progression to Alzheimer Dementia in Patients with Mild Cognitive Impairment	The phosphorylated C-terminal thrombin cleavage fragment of OPN is a biomarker of MCI that progresses to AD and is also associated with increased inflammation and gliosis.
Wirths (2010) [21]	Inflammatory changes are tightly associated with neurodegeneration in the brain and spinal cord of the APP/PS1KI mouse model of Alzheimer’s Disease (mouse study)	Identifies OPN as a marker of astrocyte and microglial activation and inflammation.
Sun (2013) [116]	Elevated osteopontin levels in mild cognitive impairment and Alzheimer’s disease	Elevated OPN levels in CSF are increased in patients with mild cognitive impairment (MCI) that progress to AD, but not in patients with stable MCI.
Rentsendorj (2018) [95]	A novel role for osteopontin in macrophage-mediated amyloid-β clearance in Alzheimer’s models (mouse study)	Elevated OPN expression correlated with protective, anti-inflammatory phenotypes of systemic macrophages that accelerated Aβ fibril clearance.
Wang (2020) [133]	Anti-human TREM2 induces microglia proliferation and reduces pathology in an Alzheimer’s Disease model (mouse study)	Antibody-mediated activation of the TREM2 receptor decreases Aβ-induced pathology while being associated with a proliferation of protective microglia and downregulation of OPN.
Quan (2021) [135]	Haploinsufficiency of microglial *MyD88* ameliorates Alzheimer’s pathology and vascular disorders in APP/PS1-transgenic mice. (mouse study)	Heterozygous deletion of *MyD88* gene decreases cerebral Aβ load and improves cognitive function, while being associated with the upregulation of microglial OPN.
De Schepper (2023) [121]	Perivascular cells induce microglial phagocytic states and synaptic engulfment via *SPP1* in mouse models of Alzheimer’s disease (mouse study)	OPN secreted by hippocampal perivascular macrophages (PVM), the primary responders to toxic agents and pathogens that cross the BBB, transformed microglia into phagocytic cells that engulf synapses in ADtg mice. This may represent phagocytic pathways that become self-destructive and drive AD progression.
Qiu (2023) [139]	Definition of the contribution of an Osteopontin-producing CD11c(+) microglial subset to Alzheimer’s disease (mouse study)	Identifies a subset of CD11+ microglia that are the sole source of OPN in ADtg mice: Protective CD11c+OPN− subset; Pathogenic CD11c+OPN+ subset.

## 7. Conclusions, Perspectives, and Emerging Treatment Options

The presence of multiple isoforms of microglia with protective and disease-associated phenotypes, and the property of microglial subtypes to switch phenotypes depending on environmental/cellular cues, suggest that OPN directs pro-inflammatory or anti-inflammatory signals depending on the myeloid and/or microglial cells with which it interacts, and with the disease stage, and context of the targeted cells. The neuroprotective actions of OPN-associated bone marrow leukocytes described by Rentsendorj et al. [95] associated with advanced Aβ plaque deposits suggest that OPN confers neuroprotective and/or reparative functions via systemic immunity at late-stage AD when neuronal damage is established and ongoing. The work is consistent with previous reports that macrophages derived from the BM phagocytose Aβ plaques more efficiently than resident microglia [141,142,143]. However, OPN-mediated neurotoxicity by multiple pathways involving microglia appears to predominate via pathways that include the enhanced formation and toxicity of Aβ plaques and NFTs, increased activities of APP cleavage enzymes, interference with Aβ clearance, and the delivery of proinflammatory cytokines. OPN can promote toxic tau accumulation and NFT fibers and contribute to disrupting the BBB, allowing the access of additional inflammatory mediators, immune cells, or pathogens to enter the brain [144]. Recently, the interrogation of human iPSC-derived microglia by CRISPR-based functional genomics identified an SPP1+ microglial subtype with a neurodegenerative disease-specific state that was targeted and neutralized by PLX3397, a selective inhibitor of colony-stimulating factor-1 [101]. PLX3397 was previously shown to be protective in neurodegenerative mouse models [145,146]. In another cell-based therapeutic approach, human neural-crest-derived turbinate stem cells were shown to neutralize Aβ neurotoxicity by suppressing OPN expression in a human cerebral organoid model of AD and in ADtg mice [147]. Thus *OPN/Spp1* is considered to be a negative DAM-associated gene and AD therapeutic target. However, because of possible OPN-directed protection mediated by systemic immunity, the outcome of OPN blockade in late-stage AD cannot be predicted.

The Aβ/tau pathway has stood the tests of time and remains the predominant hypothesis for AD progression and a promising pharmacological target, as evidenced by the recent FDA approval of two monoclonal antibodies (mAb), aducanumab and lecanemab, that target Aβ plaques and soluble AβO protofibrils, respectively [148,149]. Donanemab, an antibody that targets a modified form of deposited Aβ, is expected to receive FDA approval by the end of 2023 [12,150]. Tau antisense oligonucleotides currently in a phase 1b trial appear promising [151]. Despite this, and in part because of uncertainty over the extent of the therapeutic efficacy of Aβ/tau biologics to combat AD [94,152], a large proportion of AD drug development efforts are currently directed towards upstream and downstream targets of Aβ/tau, at accessory molecules that regulate production, activation and/or clearance, tau phosphorylation, and NFT spreading (reviewed in [153]). These include but are not limited to apolipoprotein (ApoE) pathways, neurotransmitter receptors, inflammation, oxidative stress, metabolism, vascular factors, growth factors and hormones, synaptic plasticity, and neuroprotection [153,154].

In the anti-inflammation category, at least 21 drugs are currently in phase 2 or 3 clinical trials, including small molecule and antibody inhibitors of signaling pathway intermediates/receptors (c-Kit, JAK-STAT, Src, P38-MAPK, mTor, GM-CSF, CD38, leukotriene receptor) and inflammatory cytokines (TNFα, IL-1, IL-6, and IL-12) (reviewed in [153]). Microglial target drugs include AL002, a pharmacological activator of TREM2 that is expected to slow AD progression by stimulating anti-inflammatory microglia [133]. Senicapoc is an inhibitor of the potassium calcium-activated channel subfamily N member (KCa3.1) that is a regulator of the activation, migration, and proliferation of T-lymphocytes, microglia, and macrophages. KCa3.1 is increased in the brains of AD patients and senicapoc is expected to attenuate Aβ deposition, neuroinflammation, and AD pathology [155]. TB006 is a humanized monoclonal antibody that targets galectin-3 (Gal3), a TREM2 ligand that is upregulated in patients with AD and ADtg mice. Gal3 is a key regulator of TREM2-mediated microglial activation by fibrillar Aβ [156] and TB006 is expected to reduce inflammation, decrease Aβ load, and improve cognitive behavior (NCT05476783).

In the lipid category, an open-label Phase 1/2 trial will test LX1001, an AAV vector that expresses human APOE2 delivered to patients via the spinal canal. Overexpressed APOE2 is expected to antagonize neuronal APOE4 (NCT05400330). Obicetrapib is a cholesteryl ester transfer protein inhibitor that reduces LDL in the presence of a statin. Epidemiologic studies suggest that statins decrease the risk of AD by reducing the cholesterol interactions that exacerbate neurodegeneration [157]. Obicetrapib is currently in a Phase 2 open-label exploratory proof-of-concept clinical trial (NCT05161715). Following the compelling preclinical results described above [101,139,147], further testing of antibody, stem cell, and small molecule therapies targeting OPN+ microglia are anticipated. The links between Aβ, lipid regulators, and inflammation extend beyond AD to other age related conditions including MS [140] and cardiovascular disease [158], and other work, including from the author’s group, implicates OPN related pathways and LDLR in the pathologies of renal disease and heart failure [159,160], such that OPN is a potential target for multiple conditions involving ageing, lipid dysregulation, and Aβ-related vascular inflammation.

## Figures and Tables

**Figure 1 biomedicines-11-03232-f001:**
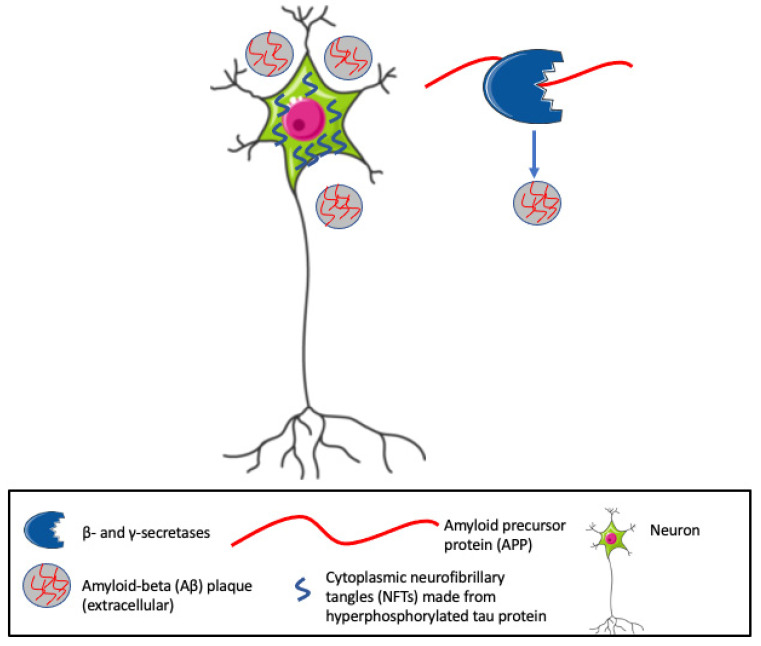
Hallmarks of Alzheimer’s Disease (AD), include cytoplasmic neurofibrillary tangles (NFTs) and extracellular amyloid beta (Aβ) plaques. β- and γ-secretases cleave the amyloid precursor protein (APP) into Aβ fragments that, along with hyperphosphorylated tau, generate neurotoxic aggregates and eventually insoluble plaques.

**Figure 5 biomedicines-11-03232-f005:**
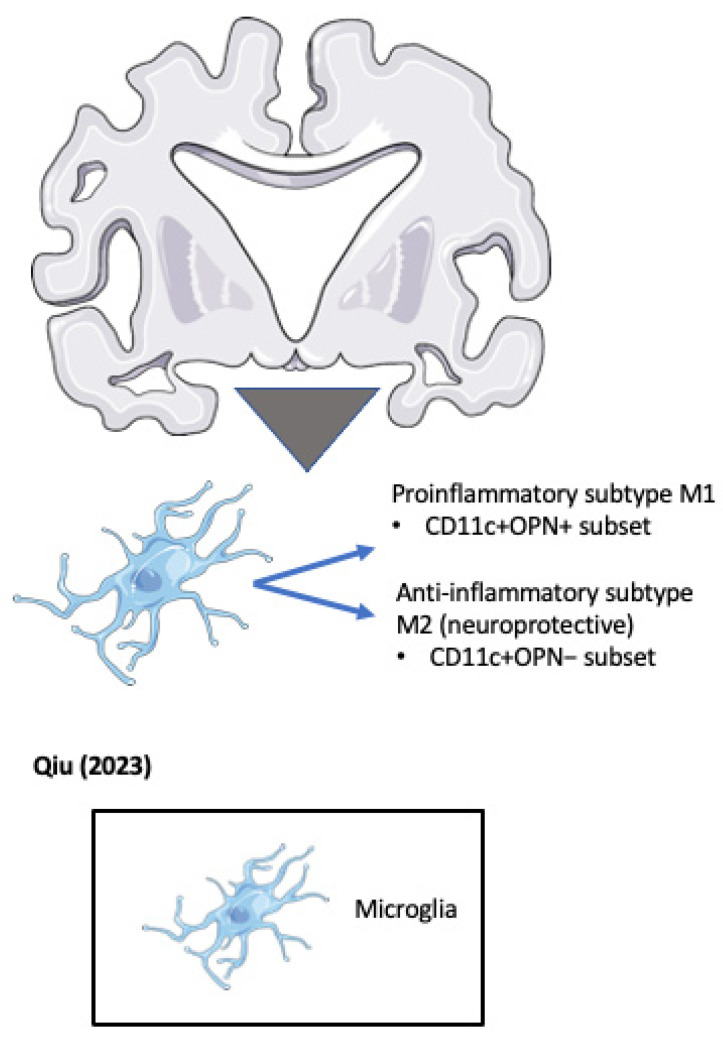
OPN positive and negative microglial subsets accumulate around pathogenic plaques in the Alzheimer’s disease (AD) brain (Qiu et al. 2023) [139].

## Data Availability

Not applicable.

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
