# Peer review of "Contextualizing the Role of Osteopontin in the Inflammatory Responses of Alzheimer’s Disease"

_biomedicines, 2023, doi:10.3390/biomedicines11123232_

Round 1

Reviewer 1 Report

Comments and Suggestions for Authors

In this review, Lalwani and co-workers gave a comprehensive view on the roles of osteopontin in Alzheimer’ disease and its possible therapeutic potential. Although the topic is of interest, there are some questions/concerns.

Some figures or schemes illustrating the structure, functions and possible roles of osteopontin would be useful.

Osteopontin (OPN) is a multifunctional protein, and its level may be increased in various disorders (lines 253-256). Moreover, “In a manner that parallels the context-dependent, opposing actions of microglial subtypes, different studies report that OPN can instigate context dependent inflammatory damage or neuroprotection and repair”.  “Therefore, the actions of OPN depend on the context, cell types, stage of disease and prevailing biochemical interactions.” Therefore, it seems to be a challenging target concerning either its specificity to AD or whether it should be inhibited or activated. Why is it of significance concerning the other possible targets?

The authors used, indeed, highly relevant literature for this review. However, a search on PUBMED using osteopontin and Alzheimer’s as key words resulted in 46 articles, 32 were published 2018-2023. The authors only reviewed 4 articles published 2018-2023. Further articles of interest may be included as well (such as Drager et al. PMID: 35953545; Lim et al. PMID: 35326480).

Please, pay attention to the references. No numbered reference should be listed in the abstract.

line 193: “via the PtdSer lig- and bridges”. Please, check this.

Reviewer 2 Report

Comments and Suggestions for Authors

Compared to people with normal cognitive function, people with Alzheimer's had, on average, three times as much osteopontin—and the microglia that produce it—in their brains. Higher levels of osteopontin correlated with greater severity of dementia.

The manuscript is not well written and introduces the topic of new research but needs to improve in several areas; these issues must be addressed to make it attractive for the journal readers.

Point 1: There is a need to revise the abstract according to the journal format.

Point 2: Authors must address the field's specific research gap filled by research.

Point 3: There is a need to clarify the central question addressed by the RA. 

Point 4: What new insights into the topic does your review provide?

Point 5:Line 19-21 Correct and clarify the sentence " Aβ and tau complexes spread through interconnected regions of the brain and bind pattern recognition receptors on microglia and astroglia to trigger inflammation and neurotoxicity that ultimately lead to neurodegeneration and clinical AD."

Point 6: It is challenging to understand the abstract of the readers.

Point 7: There is a need to revise the introduction as per the Biomedicnes.

Point 8: The writing style could be more impressive for the readers.

Point 9: Reference styles should be revised as per the format of Biomedicines.

Point 10: Major editing of the English language is required.

Comments on the Quality of English Language

Major editing of the English language is required.

Reviewer 3 Report

Comments and Suggestions for Authors

 O_s_t_e_o_p_o_n_t_i_n_ _i_n_ _A_l_z_h_e_i_m_e_r_’s_ _D_i_s_e_a_s_e_ _

This manuscript, entitled “Osteopontin in Alzheimer’s Disease”, has s several shortcomings:

1.     The title needs to be improved and made clearer and more compelling.

2.     The abstract contains references and refers to a Table. This is not appropriate and needs to be corrected.

3.     Based on the detailed studies of Braak and colleagues, the spread of plaques across the brain may be random, but the spread of tangles is not. The abstract therefore needs to be corrected.

4.     Lines 55-56: it is not correct to call hypotheses hallmarks.

5.     Lines 71: Neuropathological evidence indicates that AD begins in the entorhinal cortex with tangle formation, so the text is not correct.

6.     Lines 88: Reference 32 does not contain AD data so is not relevant and, thus, misleading.

7.     Lines 103-124: The needs to be clearer as it is only true for autosomal dominant forms of AD, which are rare and frontotemporal dementia is distinct from AD.

8.     OPN needs to be introduced earlier in the manuscript.

9.     The Table is not informative and needs to be reconfigured to present the most solid data. This may be ref 36. In which case, this undermines the argument that OPN reflects neuroinflammation as this manuscript shows an increase in OPN concentrations within pyramidal cells. This needs to be considered and discussed.

10.  The case for OPN playing a role in the pathophysiology of AD is less than convincing.

Comments on the Quality of English Language

Minor improvements needed

Round 2

Reviewer 2 Report

Comments and Suggestions for Authors

Most of the suggestions were incorporated in the revise version of the manuscript.

Author Response

Thanks.

Reviewer 3 Report

Comments and Suggestions for Authors

The manuscript is much improved. My remaing concern is Table 1, which has a title that contains the words "Alzheimer's disease", even though most of the studies referred are based on model systems. Models of AD and AD itself are not the same and so the title of this Table is incorrect. '

Comments on the Quality of English Language

fine

Author Response

We added "Studies" to the title of the Table: "Chronologic List of Publications on Osteopontin in Alzheimer’s Disease Studies."

We further corrected few typos and grammatical errors. 

Thank you.